# Empirical evaluation of the presence of a label containing standard drinks on pour accuracy among US college students

Eric Brunk[1]◉, Mark W. Becker[2]◉, Laura Bix◉[1]◉*

1 Healthcare, Universal Design and Biomechanics Lab, Packaging HUB, Michigan State University, East Lansing, MI, United States of America, 2 Department of Psychology, Cognition and Cognitive Neuroscience Group, Michigan State University, East Lansing, MI, United States of America

◉ These authors contributed equally to this work.

* bixlaura@msu.edu

**Data Availability Statement:** All relevant data are within the paper and its Supporting Information files.

## Abstract

### Purpose

Alcohol concentration has traditionally been labeled in the form of alcohol by volume (ABV). This format can cause difficulty in evaluating accuracy of a pour because it doesn't directly connect with recommendations related to "standard drinks," the approach used by the US CDC and others organizations which intend to facilitate responsible drinking behaviors. Strategies which more directly connect guidelines related to healthy drinking behaviors to alcohol labeling are needed.

### Objective

Assess how a label identifying the number of standard drinks per container impacts the ability of undergraduate students to accurately pour a standard drink.

### Design

This study employed a 3 x 2 x 2 experimental design. Undergraduates were asked to pour a standard drink from mock products from three alcohol categories (beer, wine and liquor); products were presented in two types of label (traditional ABV vs. standard drinks/container) at two concentrations of alcohol content (high and low).

### Results

We calculated standardized pour errors (pour errors in standard drink units). Analysis of these standardized pour errors suggested that 1) people tended to underpour beverages of low concentration across product categories and overpour those high in concentration. 2) When the standard drink label was present, pour accuracy was improved, when compared with pours from containers affixed with ABV labels in low alcohol concentrations across all product categories (beer, wine and liquor). 3) For treatments that comprised high concentrations of alcohol, the standard drink label significantly increased accuracy only for beer. However, it is worth noting that beer with an ABV label was the condition with the most dramatic

**Funding:** The study presented herein was funded internally partially through proceeds developed by a conference our laboratory holds every other year, and partially through a grant from the College of Agriculture and Natural Resources that intends to enable Undergraduate Students to engage in research projects, provided to Eric Brunk. As mentioned above, a portion of this money remained upon completion of his UG Degree and, as such, was carried forward to support his MS work. A portion of Dr. Bix's salary is covered through the USDA Hatch Act Program within NIFA under project # MICL 00263. The funders had no role in study design, data collection and analysis, decision to publish, or preparation of the manuscript.

**Competing interests:** The authors have declared that no competing interests exist.

**Abbreviations:** ABV, Alcohol by Volume; ANOVA, Analysis of Variance; BOPP, Biaxially Oriented Polypropylene; CANR, College of Agriculture and Natural Resources; CAS, College of Communication Arts and Sciences; CDC, Centers for Disease Control and Prevention; HUB, Healthcare, Universal Design, and Biomechanics; NIAAA, National institute on Alcohol Abuse and alcoholism; PDP, Principle Display Panel; TTB, Alcohol and Tobacco Tax and trade Bureau.

overpours, and these problematic overpours were dramatically reduced by the addition of a standard drink label.

## Conclusions

Our work empirically supports the notion that Undergraduate students are better able to accurately assess and pour a standard drink of alcohol from bottles incorporating a label which includes standard drinks/container vs. those with traditional ABV labeling. That said, the effect is quite different for each alcohol category: beer, wine, and liquor and depends on whether the product is high or low in concentration of alcohol for its category; as such, policy makers should consider alcohol categories and concentrations from a public health perspective when recommending changes to labeling.

## Introduction and background

The US government mandates specific information content and formatting for the labels of many consumer products, including: foods, over-the-counter medications (OTC), and alcohol. Regulations related to OTC and food labels are promulgated by the US Food and Drug Administration (FDA), an Agency within the Department of Health and Human Services (DHHS). By contrast, the labeling of alcohol is regulated by the Alcohol and Tobacco Tax and Trade Bureau (TTB), an Agency within the US Department of Treasury. Differing missions within the Departments, and the Agencies that they oversee, result in unique motivations which express themselves in the form of the labels that they require across product categories.

While food and drug labeling is intended to motivate informed, healthful choices by consumers, alcohol labeling, with roots in the abrogation of prohibition, has tended to focus on ensuring that consumers receive what they believe that they purchased (type, origin and content) and that producers pay appropriate taxes on the products that they sell. As such, the labeling of alcohol content has been traditionally expressed as alcohol by volume (ABV) or proof (2x ABV), measures which don't easily lend themselves to informed decision-making for consumption. This is despite the fact that most guidance related to (un)/healthy drinking behaviors, in the US and throughout the world, are not defined on the basis of ABV, but in "standard drinks" (SD) (see Table 1) [1]. A standard drink is defined as a drink containing 14 g (0.6 fl oz) of pure alcohol. The standard drink concept has been indicated as "foundational knowledge" for understanding health policy aimed at reducing harm from alcohol [2]. It is an attractive approach to those lobbying more comprehensive labels for consumer products (food and drug) and also appeals to health policy makers because it accounts for both the strength and the volume of a beverage, allowing for comparisons within and across beverage categories [2].

**Table 1. Standard drinking behaviors as recommended by the National Institute on Alcohol Abuse and Alcoholism (NIAAA) [1].**

| Drinking Behavior | Definition | For Men | For Women |
|---|---|---|---|
| Moderate Drinking | Standard drinking behavior | 2 standard drinks per day | 1 standard drink per day |
| Binge Drinking | Pattern of drinking that brings blood alcohol concentration (BAC) levels to 0.08 g/dL | 5 standard drinks in 2 hours | 4 standard drinks in 2 hours |
| Heavy/Excessive Alcohol Use | Pattern of excessive binge drinking | 4+ standard drinks on any given day | 3+ standard drinks on any given day |

The disconnect between the labeling standards and consumption recommendations becomes painfully apparent in the following example. Consider what is required to determine the number of standard drinks in a 10 oz glass of wine which is labeled as having 14% ABV. The formula for conversion requires one to convert oz into grams (10 oz X 29.574 g/oz = 295.74g), then multiply that result by the ABV (295.74 g*(14/100) = 41.4), subsequently one must multiply by the specific gravity of alcohol (~0.7936 x 41.4 g = 32.858 g) to determine that there are 32.858 grams of pure alcohol per 10 oz glass. Finally, to convert this to standard drinks one would have to know that a standard drink is equivalent to consuming 14 grams of pure alcohol (see Table 1); as such, you divide the number (32.858 g) by 14 g/standard drink, to discover that the 10 oz glass of wine is the equivalent of 2.3 standard drinks.

Clearly, the conversion of ABV to standard drinks is not intuitive. As a result, the National Institute on Alcohol Abuse and Alcoholism (NIAAA) has promoted more intuitive estimates to guide consumers. NIAAA suggests estimating a standard drink as equivalent to approximately 12 oz of beer, 8–9 oz of malt liquor, 5 oz of wine, and 1.5 oz of liquor [3]. These estimates are based on the assumption of 5%, 7%, 12%, and 40% ABV for each drink type, respectively. Although this simplifies things considerably, it disregards the wide variation that can be present in ABV for a single category of alcohol, particularly beer and liquor. Consider 12 oz cans of Budweiser (5% ABV), and Sam Adams's Triple Bock (17.5% ABV). While the NIAA heuristic (a 12 oz can of beer is a standard drink) is true for the Budweiser, the Triple Bock is equivalent to 3.5 standard drinks, yielding the potential for significant over-consumption; the consumer may be getting 2.5 more drinks than they interpreted from a single pour.

College students and young adults are frequently the focus of consumption studies because they represent a specific, at-risk group. Over-consumption can be particularly problematic for this cohort, which is susceptible to binge drinking [4] and resultant tribulations. In the US, binge drinking has become a serious problem, especially among those aged 18 to 25 [5]; one study found that a large portion of college students drink at peak levels well beyond the binge threshold (see Table 1) [6].

Although we recognize that decision making is multifactorial in nature and no single strategy can eliminate overconsumption behaviors, we were interested in whether or not a change in the US approach to labelling could better equip people motivated to consume appropriately to do so than the current labelling standard (ABV).

Research supports the idea that drinkers have difficulty identifying a serving size as it relates to the amount of alcohol they choose to consume. One study of U.S college students asked participants to complete an alcohol survey and pour a variety of alcoholic beverages to a volume they deemed to be a standard drink. The study found that the students did not know how to define standard drinks accurately and overestimated volume when pouring [7]. Other studies of young-adult drinkers and college students have also surmised that they are generally unaware of the alcohol content in beverages, as well as national recommendations regarding responsible drinking behaviors [8–10]. A study focusing on US college students and bartenders, suggests that both tend to over-pour liquor when utilizing shot glasses [11].

Revising the labelling of alcohol content to be more aligned with recommendations for consumption (i.e. standard drinks) (see Table 1) represents one change with the potential to enable informed decisions related to drinking behaviors. Even so, recent US regulatory action in the form of a proposed rule focused on alcohol labeling entitled, "Modernization of the Labeling and Advertising Regulations for Wine, Distilled Spirits, and Malt Beverages" [12], allows for, but does not mandate, such a change. The proposed rule (§ 7.65(b)(1)) states, "Other truthful, accurate, and specific, factual representations of alcohol content, such as alcohol by weight, may be made, as long as they appear together with, and as part of, the statement of alcohol content as a percentage of alcohol by volume [ABV]." Although other (truthful)

information is allowed related to alcohol content, only the ABV remains as a requirement. As a result, the rule, as proposed, has been criticized as falling "dramatically short" of what is needed to "modernize" alcohol labeling [13]. The criticism, levied by three national consumer advocacy groups, indicates that the rule fails to require uniform disclosure of key information required for consumers to make informed choices for purchase and consumption of these products (e.g. alcohol content, serving size, calories, ingredients and allergen information).

Here, we focus specifically on the presentation of alcohol content, with the goal of investigating whether presenting information about the number of standard drinks per container enables young adults to make more accurate assessments of the serving size which constitutes one standard drink. A systematic review of the peer-reviewed and grey literature published between 1990 and 2016 concludes that overall knowledge related to standard drinks is low, but that standard drink label use can assist consumers to more accurately identify and pour a serving [14]; however, none of the published studies were conducted with US consumers. That said, studies conducted in Canada [15, 16], Australia [17, 18] and Europe were consistent with regard to these conclusions, and Australia has since adopted a standard drink labeling practice for alcoholic beverages.

We tested a standard drink label positioned on the front of the package using US college students, to see if this labeling strategy enabled young adults to better assess the quantity of alcohol comprising a standard drink as compared with the current US labelling standard (ABV). Additionally, we postulated that product category (beer, wine and liquor) and alcohol concentration (high vs low) had the potential to impact pour volumes assessed by young adults.

## Materials and methods

### Recruitment and consent

We based our power calculation on the most conservative comparison performed; a single sample t-test that compared a particular pour condition's mean pour error to zero. Results suggested that to achieve the power to detect a moderate effect size of with power of 0.95 required 84 participants [19, 20]. Ultimately, we recruited and tested 84 undergraduate participants in accordance with methods approved by the Institutional Review Board (IRB) at Michigan State University as STUDY0000083.

Recruitment was conducted using the student subject pool administered by the College of Communication Arts and Sciences at Michigan State University. To participate subjects had to be: 18 or older, enrolled as a college student, have transportation to the study site (a campus laboratory) and be willing to share contact information for the purpose of scheduling. All subjects provided informed, written consent, and were paid $20 in exchange for their time.

### Study stimulus

This study utilized a 3 x 2 x 2 experimental design, with 3 alcohol categories (beer, wine and liquor), 2 label types (ABV and standard drinks/container), and 2 alcohol concentrations (high and low in each of the alcohol categories- See Table 2). This resulted in a total of 12 different labels (3 categories x 2 label types x 2 concentrations of alcohol). For each alcohol category (beer, wine, and liquor), one original brand was created and used for each of the label types and alcohol concentrations (See Fig 1 for an example); information related to the alcohol concentration was positioned in the lower right corner of the label and the bottle capacity in the lower left and presented in accordance with current US regulation. The standard drinks per container were calculated using the standard drink calculator available at "Rethink Drinking" published by the NIAA [21]. Table 2 presents information related to stimulus design, pouring container and receiving container, which varied by alcohol category.

**Table 2. Summary of stimulus design.**

| Category | Container | Container Size* | Receptacle | Receptacle Size* | Brand | Alcohol concentration | Standard Label (Alc. By Vol.) | Standard Drinks per container | Standard Drink Size* |
|---|---|---|---|---|---|---|---|---|---|
| Beer | Bottle | 12 fl oz | Solo cup | 16 oz | "High Seas Brewing Co." | Low | **5%** | **1** | 12 fl oz |
| | | | | | | | **(A)** | **(B)** | 355 mL |
| | | 355 mL | | 473 mL | | High | **10%** | **2** | 6 fl oz |
| | | | | | | | **(C)** | **(D)** | 177 mL |
| Wine | Bottle | 25.4 fl oz | Stemless wine glass | 14 oz | "Simply Divine Fine Wines" | Low | **10%** | **4.2** | 6 fl oz |
| | | | | | | | **(E)** | **(F)** | 177 mL |
| | | 750 mL | | 414 mL | | High | **15%** | **6.3** | 4 fl oz |
| | | | | | | | **(G)** | **(H)** | 118 mL |
| Liquor | Bottle | 25.4 fl oz | Bar shot glass | 9 oz | "Caribbean Dreams" | Low | **20%** | **8.4** | 3 fl oz |
| | | | | | | | **(I)** | **(J)** | 89 mL |
| | | 750 mL | | 266 mL | | High | **40%** | **16.9** | 1.5 fl oz |
| | | | | | | | **(K)** | **(L)** | 44 mL |

Bold font indicates the 12 different label variations used in the study. The ABV and standard drink labels indicate the same amount of alcohol per container in low and high formats. Each letter corresponds to the stimulus labels located in Table 2.

*Volume provided in fl oz and mL due to differences in labeling requirements among alcoholic beverage categories (beer, wine, liquor).

## Data collection

Participants began with a brief, computer-based questionnaire related to their knowledge of consumption recommendations, as well as their drinking behaviors. Once complete, each participant engaged in a series of 24 pouring tasks, in two blocks of 12 unique treatments. They were instructed, "Pour a standard drink into the glass in front of you. Feel free to use any information present on the bottle to help inform your pour" (see Table 2 for bottles and glasses by treatment type). Order of presentation was randomized for each participant within the first block of 12. This order was repeated for the second block of twelve, comprised of a single replicate of each treatment. The volume of liquid each subject poured was measured (post-hoc) for every trial.

## Data preparation & analyses

Each participant produced two pours within each condition of the 3x2x2 design. We began by averaging the volume (ml) of the two pours within a condition for each subject, producing a

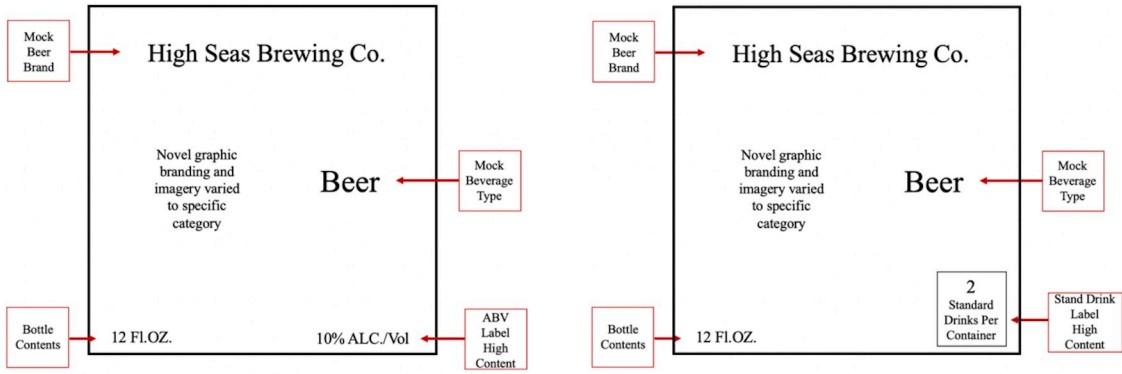

**Fig 1. Example stimulus design.**

**Table 3. Summary of variables.**

| Variable | Explanation |
|---|---|
| Observed Pour | Participant pour recorded (in mL) during experiment as measured using a graduated cylinder with a capacity of 250 mL or 500 mL, and a sensitivity of 2 mL and 5 mL, respectively |
| Mean Pour Volume (MPV) | Average of the volume (in mL) of the two pours that a participant made within a condition. |
| Accurate Volume (AV) | The volume (in mL) that would accurately represent a standard drink for a given condition, as calculated using NIAA's drink calculator [21]. |
| Mean Standardized Pour Error (SPE) | Each subject's pour error for a condition expressed in standard drink units. The formula to calculate it is SPE = (MPV-AV)/AV, where the MPV and AV values are for a specific condition. |

"mean pour volume" dependent variable (see Table 3). Six participants produced mean pour volumes for at least one condition that were more than three standard deviations from the overall mean for that condition. Data from those participants were eliminated from further analysis, resulting in analysis that included data collected from 78 participants.

We also converted the mean pour volume measure for each subject into a "standardized pour error" (SPE) variable (see Table 3). To do this we subtracted the "accurate volume" representing a standard drink (AV) for a given condition from each subject's mean pour volume (MPV) for that condition, and divided the result by the accurate volume representing a standard drink (AV). Mathematically, this conversion is SPE = (MPV-AV)/AV, converting the pour errors into standard drink units. The rationale for transforming data into standard drink units is that it is the most relevant metric from the standpoint of public policy, is how much alcohol a person is pouring as opposed to the volume of the pour. That is, from a health perspective overpouring 30 milliliters of liquor is more concerning that overpouring 30 milliliters of beer. Once converted to standardize pour errors (SPEs), an overpour of 0.5 SPEs for beer and an overpour of 0.5 SPEs for liquor are equivalent in terms of the amount of alcohol that was overpoured; in both cases the pour was equivalent to 1.5 standard drinks or 21 grams of pure alcohol for each pour. Finally, we note that given the method of subtraction, positive standardize pour errors represent overpours, while negative values represent underpours.

An initial review of the data revealed them to be non-normally distributed; attempts to transform the data via standard transformations (e.g., log, double log) did not produce normal distribution. That said, we note that the ANOVA is fairly robust to violations of non-normality [22]. Given this robustness, we performed ANOVA omnibus tests, but in follow-up analyses, performed both paired t-tests and Wilcoxon Signed Rank Tests, a non-parametric analysis that does not have an assumption of normality. All statistical analyses were performed using SPSS [23].

## Results

Data collected from our survey support previous findings utilizing college-aged participants which conclude them to be poorly acquainted with the idea of standard drinks [7–10]. This was true of even questions which utilized simple heuristics published by the NIAA. Specifically, only 60% (n = 50) of our respondents were able to correctly identify 12 oz of beer at 5% ABV as a standard serving. This dropped to 27% (n = 23) for wine (5 oz of wine at 12% ABV) and 17% (n = 14) for liquor (1.5 oz of liquor at 40% ABV).

The average mean pour volumes (MPV) by treatment are presented in Fig 2 juxtaposed against the volume representing the volume of an accurate standard drink (AV) for each treatment (presented in black). While we present these mean pour volumes for completeness, our statistical analyses were performed on the standardize pour errors (SPE).

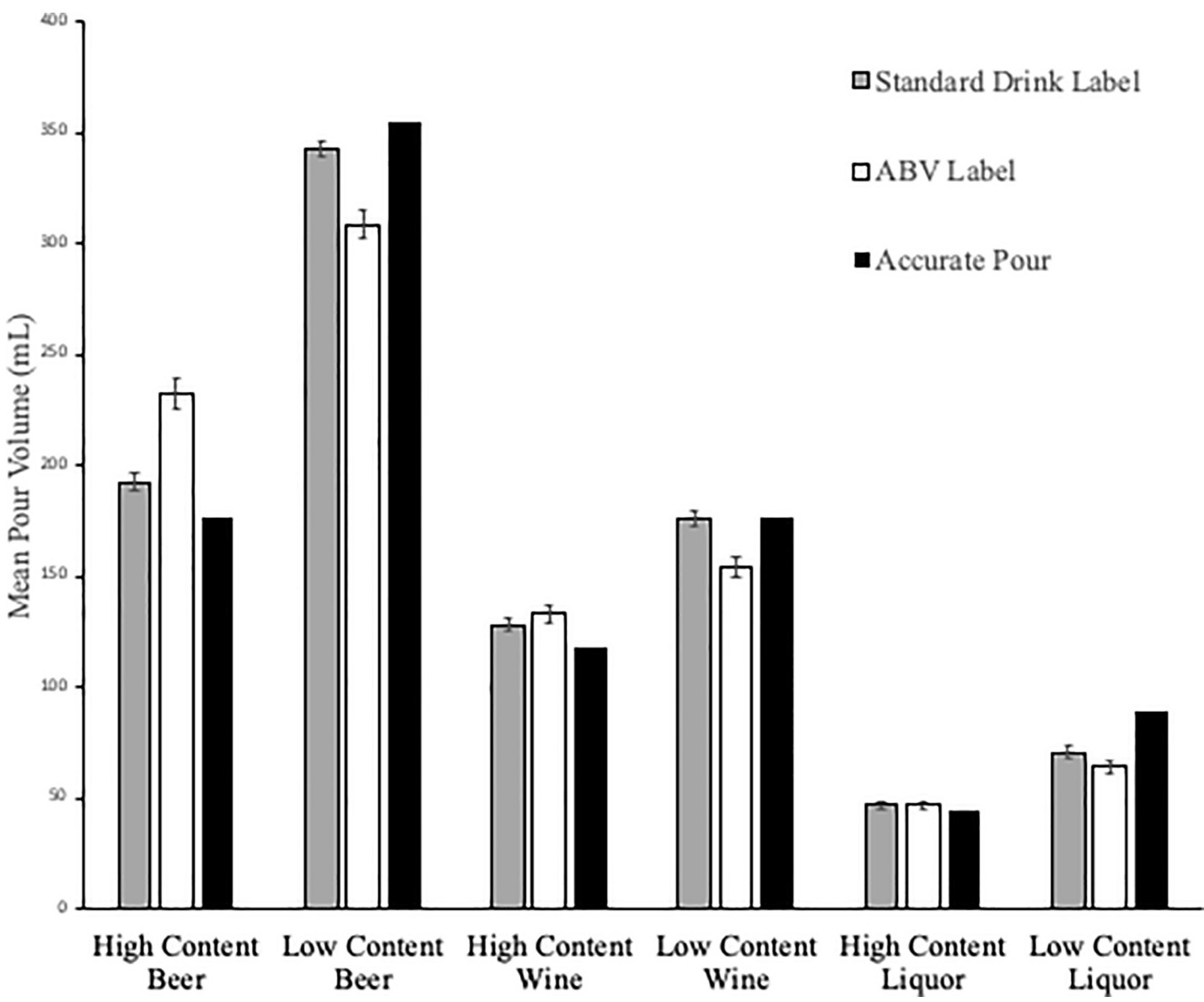

**Fig 2. Mean pour volume by category, label type, and alcohol concentration level.**

The mean standardized pour errors across participants are presented in Fig 3. An omnibus 3 (alcohol category: Beer, Wine, Liquor) x 2 (concentration: high and low) x 2 (label type: ABV and standard drink label) within subjects ANOVA on the standardize pour errors revealed significant main effects of alcohol category $F_{(2, 154)} = 13.35$, $p < 0.001$, $\eta_p^2 = 0.148$, and concentration, $F_{(1,77)} = 194.67$, $p < 0.001$, $\eta_p^2 = 0.717$. While the main effect of label type did not approach significance, $F_{(1, 77)} = 0.681$, $\eta_p^2 = 0.002$, it was qualified by significant two way interactions of alcohol category x alcohol concentration, $F_{(2, 154)} = 13.97$, $p < 0.001$, $\eta_p^2 = 0.16$; alcohol category x label type, $F_{(2, 77)} = 7.93$, $p < 0.001$, $\eta_p^2 = 0.093$; and alcohol concentration by label type, $F_{(1, 77)} = 51.91$, $p < 0.001$, $\eta_p^2 = 0.40$. The three-way interaction was also significant, $F_{(2, 154)} = 9.87$, $p < 0.001$, $\eta_p^2 = 0.114$. One clear source of these interactions was that the direction of mean standardize pour errors differed as a function of alcohol concentration, with consistent overpours for high concentration conditions, and consistent underpours

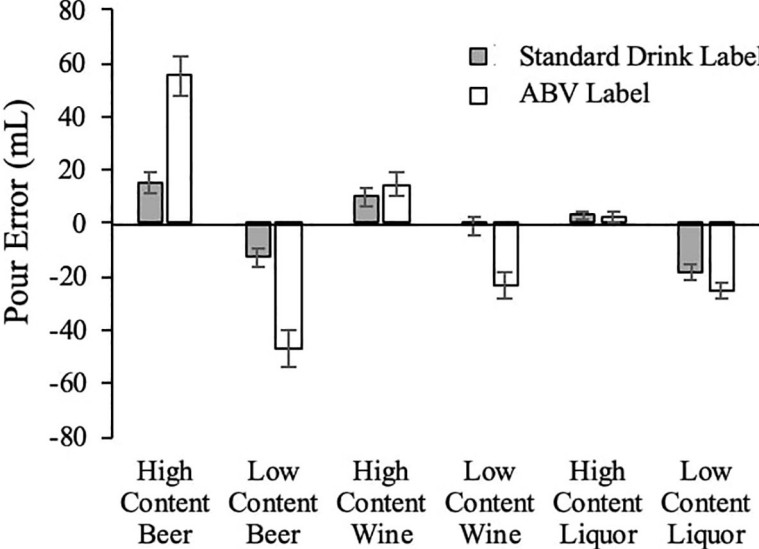

**Fig 3. Mean standardized pour error by condition.**

for low concentration conditions (see Fig 3). Thus, to unpack the three-way interaction we ran two 3 (alcohol category) x 2 (label type) ANOVAs–one for high concentrations and one for low concentration conditions. We believe that analyzing the high and low conditions separately is ideal since the types of errors for the two conditions are qualitatively distinct with different health consequences (over pouring has the potential to lead to adverse effects where underpouring carries less risk).

The pattern of data with for the high content conditions revealed significant main effects of alcohol category, $F(2, 154) = 6.693$, $\eta_p^2 = 0.080$, $p = .002$, label type, $F(1, 77) = 11.915$, $\eta_p^2 = 0.134$, $p = 0.001$, with a significant interaction related to alcohol category x label type, $F(1, 77) = 12.128$, $\eta_p^2 = 0.136$, $p<0.001$. The cause for all three significant results appears to be that there were very high error rates for beer labeled with the ABV label, and these errors were mitigated by the use of a standard drink label. To examine this issue further, for each alcohol category, planned within subjects tests comparing SPEs between the standard drink labels and ABV labels were performed. These tests reveal that pours were significantly more accurate with the standard drink label for beer, with no apparent significant difference in accuracy when two label types were compared for wine or liquor (see Table 4 for statistics). The

**Table 4. Paired sample t-tests and Wilcoxon signed rank tests comparing ABV to standard drink errors for each type of drink.**

|  | High Content | Low Content |
|---|---|---|
| Beer | **t = 5.07, p < .001** | **t = -5.33, p < .001** |
|  | **Wilcoxon p < .001** | **Wilcoxon p < .001** |
| Wine | t = 1.10, p = .28 | **t = -4.56, p < .001** |
|  | Wilcoxon p = .183 | **Wilcoxon p < .001** |
| Liquor | t = .26, p = .80 | **t = -2.82, p < .006** |
|  | Wilcoxon p = .90 | **Wilcoxon p = .004** |

All t-tests df. = 77. Bold values represent significant errors. Negative t-values represent underpours; positive t-values represent overpours.

**Table 5. Single sample t-tests and Wilcoxon signed rank tests.**

| | High Content | | Low Content | |
|---|---|---|---|---|
| | *ABV* | *Standard Drink* | *ABV* | *Standard Drink* |
| Beer | **t = 7.44, p<0.001** | **t = 3.66, p<0.001** | **t = -7.10, p < .001** | **t = -3.89, p < .001** |
| | **Wilcoxon p<0.001** | **Wilcoxon p = 0.005** | **Wilcoxon p < .001** | **Wilcoxon p < .001** |
| Wine | **t = 3.49, p = 0.001** | **t = 2.98, p = 0.004** | **t = -4.883, p < .001** | t = -.212, p = .833 |
| | **Wilcoxon p = 0.002** | **Wilcoxon p = 0.009** | **Wilcoxon p < .001** | Wilcoxon p = .986 |
| Liquor | t = 1.47, p = 0.15 | t = 1.73, p = 0.09 | **t = -8.91, p<0.001** | **t = -6.95, p<0.001** |
| | Wilcoxon p = 0.45 | Wilcoxon p = 0.45 | **Wilcoxon p<0.001** | **Wilcoxon p<0.001** |

Each condition's error was compared to zero. All t-tests df. = 77. Bold values represent significant errors. Negative t-values represent underpours; positive t-values represent overpours.

standard drink labels significantly reduce overpours for high content beer as compared to the traditional labeling approach (ABV). Importantly, high content beer labeled with an ABV label was the condition with the largest overpours, representing about an additional third of a standard drink (see Fig 3). By contrast, even with the ABV label, there were relatively small overpours for high content wine and liquor, with overpours of only about a tenth of a standard drink; as such, including a standard drink label yielded no evidence of a difference for these small overpours.

Turning to the low content conditions, the ANOVA suggests main effects of alcohol category, $F(2, 154) = 27.615$, $\eta_p^2 = 0.264$, $p < 0.001$, and label type, $F(1, 77) = 42.515$, $\eta_p^2 = 0.356$, $p < 0.001$, with no evidence of a significant interaction, $F(2, 154) = 1.218$, $p = 0.299$, $\eta_p^2 = 0.016$. This pattern suggests that the inclusion of the standard drink label significantly reduced errors and did so to a similar extent for each alcohol type. To confirm, we ran a set of paired t-tests and Wilcoxon Signed Rank Tests comparing ABV label pour performance to that when standard drink labels were present for each alcohol category. The standard drink label significantly increased pour accuracy for all alcohol categories (see Table 4 for statistics). Even so, it is important to keep in mind that the errors in these low content conditions represent underpours, so the improved accuracy related to the standard drink label represents pouring additional alcohol.

Finally, to determine whether these observed errors were meaningful (i.e. were significantly different from zero), we performed a series of single sample t-tests and Wilcoxon Tests comparing each condition's mean standardized pour error to zero (See Table 5). These analyses show that almost all conditions result in significant errors. Specifically, the only ABV condition that did not produce a significant error was the high content liquor condition, suggesting that participants were somewhat conservative in their pours of liquor. In addition, although the standard drink label reduced the errors in most conditions, only in the low content wine did this reduction result in a complete elimination of the error. That is, for most conditions, people continued to significantly deviate from the AV, having residual errors, even with the standard drink labels.

## Discussion

The results of this experiment suggest that providing a standard drinks label, in place of the traditional ABV, enables participants to more accurately assess a serving size that is equivalent to recommended standards based on the amount of alcohol. This conclusion is generally in line with prior reports on the efficacy of standard drink labels [11–16]; however, our results incorporate more nuances. Specifically, not only did we investigate how this labeling strategy is influenced by both the concentration of alcohol in the product and its product category, we

analyzed pour errors in units that considered the concentration of the alcohol, standard drink units. As such, results are more directly useful for those interested in health policy. Further, while the use of a standard drinks labels significantly reduced errors, there tended to be small residual errors (in the original direction of error), even in the presence of standard drink labels. That is, even with the improvement in pour accuracy acquired with the standard drink label, people continued to slightly overpour the drinks that were at the higher concentration and underpour those at the low.

One potential explanation for this pattern of over and under pouring (as a function of alcohol concentration) is that our participants have a preconceived notion of how large of a pour represents a standard drink for each alcohol category, and they adjust from that preconceived notion based on the information on the label. However, with the ABV label their adjustment is insufficient. That is, while they reduce the size of the of the pour for high concentration drinks, they under-adjust, resulting in an overpour. Similarly, when they are confronted with a low concentration, they increase the pour but fail to fully adjust, resulting in an underpour. The failure to adequately adjust from their preconceived notion of the size of a standard pour is reminiscent of the classic "Anchoring and Adjustment" heuristic [24, 25]. This heuristic suggests that people have a starting point, or anchor, that represents an initial evaluation of the value or quantity of an item, and then they adjust from that anchor based on additional information. However, a systematic bias has been noted whereby people will not adjust far enough, resulting in a final outcome that is too close to the anchor, not the end goal. While that decision-making heuristic may be at play in these judgements, our results suggest that providing a standard drink labels provides more (or better) information so that it leads to a larger adjustment from the anchor than the traditional labeling illicits.

It is also worth noting that, from a public health perspective, the benefit of the standard drink label were differential for high and low concentration examples of a drink type. While the standard drink label led to more accurate pours for all three low concentration drinks, the improvement in accuracy moved the person pouring to pour more alcohol. For the high concentration drinks, the standard drink label only produced more accurate pours than the ABV label for the beer category. Even so, it is worth noting that the high concentration beer with ABV labelling was the condition that had the largest overpours—the condition that is most problematic from a health and safety perspective. The fact that the standard drinks label greatly reduced overpours in this most troubling condition suggests that the label may be most effective in the area of the greatest public good. In light of these findings, policy makers should consider alcohol categories and concentrations from a public health perspective when recommending changes to labeling.

## Limitations

One limitation of the current study is that it remains unclear why the efficacy of the standard label differed as a function of alcohol concentration within a class. One possible explanation is that our young adult participants may have different levels of familiarity and experience with different types of alcoholic drinks and that familiarity may influence the effectiveness of the standard drink label. For instance, it appears that our subjects were somewhat cautious with liquor, only slightly overpouring in the ABV high content condition and dramatically underpouring in the ABV low content condition. This initial wariness may have limited the ability of the standard drinks to significantly moderate behavior. However, this reasoning is purely speculative and more research that directly assessed the relationship between drink familiarity and the effectiveness of a standard drink label would be required before drawing conclusions about the potential explanation.

A second limitation is that the current study was a pour study rather than a consumption study. Although our pour results are consistent with the existing literature and suggest that labeling incorporating standard drink information improves consumers' ability to identify the volume of liquid comprising a standard drink, it is unclear how this ability impacts actual consumption behavior. In fact, it has been suggested that those with the best recall of label information are, in fact, the biggest drinkers [26]; it is possible that providing accurate information has the potential to further enable high risk drinking behaviors. This notion has presented itself in historical policy. After prohibition was repealed, Congress passed a law which *banned* labeling beer with ABV [27], fearing its presence would result in "strength wars" between brewers attempting to increase demand for their product. However, in 1995 Coors successfully challenged the ban on ABV labeling for beer in the US Supreme Court [28] and it is now allowed. Even so that fear still exists, and there is recent evidence which seems supports the assumption. Following the implementation of an standard drink label in Australia [29], research found that college students were aware of these labels and reported using them to help "choose the strongest drinks for the lowest cost." Thus, it seems that, particularly for young people, providing easily accessible information about alcohol concentration has the potential to enable overindulgence more effectively.

A final limitation was that the volume of the standard drink for our low content beer condition was the full receptacle (12 ounces), limiting the ability of people to overpour in this condition. We chose this as an ecologically valid condition (most low ABV beers cannot be overpoured in realistic conditions based on bottle (and glass) sizes, and glass size has been indicated to be a contributing factor to pour volume [30–33], nonetheless, this present an experimental imperfection.

## Conclusion

Our results show clear evidence that a standard drink label is more effective than a standard ABV label at communicating accurate information about the serving size that represents a standard drink. In a majority of our conditions the standard drink label led to more accurate pours of a standard drink serving size, and in no cases did it perform significantly worse than the current ABV label. Given that a vast majority of the public health messaging around responsible drinking has focused on the standard drink metric, rather than the ABV metric, labels that allow one to better estimate the serving size that represents a standard drink are important to the efficacy of those public health campaigns. As such, we believe strongly that adopting a standard drink label, as has been done in Australia, can be an important contributor to effective public health messaging about responsible drinking. Further, providing such a label is consistent with calls of consumer groups for labelling that contains key information required for consumers to make informed choices for purchase and consumption of these products [13].

That said, the label alone will not accomplish these public health goals. An effective label can communicate accurate information about the product, but the responsible use of that information requires education and public health campaigns that convince people to utilize that information to make responsible decisions. In short, for such campaigns to be successful requires two elements. First, one must effectively convey the information that can empower consumers to make more healthy decisions. Second, the campaign must convince people to utilize that information to do so. The standard drink labels we tested here can assist with providing accurate alcohol content information and doing so is a necessary precondition for consumer to be able to make more healthy decisions. However, accurate product information alone is not sufficient to necessarily impact behavioral change. Instead this improved

information needs to be coupled with effective public health campaigns to convince people to make healthy decisions.

## Supporting information

**S1 File.**
(XLSX)

**S2 File.**
(XLSX)

## Acknowledgments

The authors would like to thank all study participants who consented to be a part of this study. The authors would also like to thank Shiva Esfahanian, Alyssa Harben, and Jiyon Lee for their help in data collection.

## Author Contributions

**Conceptualization:** Eric Brunk, Mark W. Becker, Laura Bix.

**Data curation:** Eric Brunk.

**Formal analysis:** Eric Brunk, Mark W. Becker, Laura Bix.

**Funding acquisition:** Laura Bix.

**Investigation:** Eric Brunk, Laura Bix.

**Methodology:** Eric Brunk, Mark W. Becker, Laura Bix.

**Project administration:** Laura Bix.

**Resources:** Laura Bix.

**Supervision:** Laura Bix.

**Writing – original draft:** Eric Brunk.

**Writing – review & editing:** Eric Brunk, Mark W. Becker, Laura Bix.

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
