## [Decision Letter · Decision Letter 0]

3 Aug 2020

PONE-D-20-14913

Effects of a novel labeling strategy: Does presentation of standard drinks per container impact pour accuracy of alcohol among US college students?

PLOS ONE

Dear Dr. Bix,

Thank you for submitting your manuscript to PLOS ONE. After careful consideration, we feel that it has merit but does not fully meet PLOS ONE’s publication criteria as it currently stands. Therefore, we invite you to submit a revised version of the manuscript that addresses the points raised during the review process.

Thank you for your submission to PLOS one. I apologise for the length of time this paper was under-review and thank them for their patience, it took a long time to find suitable and available reviewers. The paper makes an important contribution to the literature in terms of enhancing our understanding of beverage labelling strategies that can help improve the accuracy of pouring a standard drink (which could potentially make it easier for people to moderate their intake in relation to recommended guidelines). I am very grateful for reviewers careful consideration of the paper, its strengths and limitations and hope that their suggestions are helpful in revising the paper. One omission I noted was a limitations section, which should be added to the discussion as this will likely help guide future research in this area.

We look forward to receiving your revised manuscript.

Kind regards,

Victoria Manning

Academic Editor

PLOS ONE

Additional Editor Comments:

I apologise for the length of time this paper was under-review and thank the authors for their patience as it took a long time to find suitable reviewers. The paper makes an important contribution to the literature in terms of enhancing our understanding of beverage labelling strategies that can help improve the accuracy of pouring a standard drink (which could potentially make it easier for people to moderate their intake in relation to recommended guidelines). I am very grateful for reviewers careful consideration of the paper, its strengths and limitations and hope that their suggestions are helpful in any revisions. One omission I noted was a limitations section, which should be added to the discussion as this will likely help guide future research in this area.

Journal Requirements:

2. Please ensure that you refer to Figure 7 in your text as, if accepted, production will need this reference to link the reader to the figure.

3. Please upload a copy of Figure 9, to which you refer in your text by line 480. If the figure is no longer to be included as part of the submission please remove all reference to it within the text.

Reviewers' comments:

Reviewer's Responses to Questions

**Comments to the Author**

1. Is the manuscript technically sound, and do the data support the conclusions?

Reviewer #1: No

Reviewer #2: Yes

2. Has the statistical analysis been performed appropriately and rigorously? 

Reviewer #1: Yes

Reviewer #2: Yes

3. Have the authors made all data underlying the findings in their manuscript fully available?

Reviewer #1: Yes

Reviewer #2: Yes

4. Is the manuscript presented in an intelligible fashion and written in standard English?

Reviewer #1: No

Reviewer #2: Yes

5. Review Comments to the Author

Reviewer #1: This is an interesting study that explores the impact of standard drink labels, compared to existing ABV labels, on accuracy of drink pouring across different types of alcoholic drink. However, the manuscript is very long and in places could be edited to reduce length and improve clarity. In addition, the stated objectives and discussion of results should reflect the outcomes examined in the study (i.e., pouring accuracy) rather than consumption, and the discussion of results should be placed within the context of previous literature. Outlined below are some specific comments that may be helpful for improving clarity. The manuscript should also be checked for grammatical and typographical errors.

Introduction

• The introduction is long and could be edited to make the points more succinct. For example, the first paragraphs provide quite a lot of detail of the current regulations around alcohol content labelling that could be summarised in a few sentences.

• The introduction could also be structured more clearly. For example, it would be useful to define ‘standard drinks’ (i.e., 14g pure alcohol in US – para. 7, line 147 and para. 8, line 163) and how these are used in drinking guidelines (Table 1) when they are first mentioned (para. 3, line 110). Followed by why standard drink labelling may be preferable to ABV information, given the wide variation both within (para. 6, line 136 and para. 9) and across drink categories and sizes (para. 7, line 150).

• Para. 8, lines 164-171: The explanation of the calculation is quite long and complex, and misses out some elements (e.g., dividing g of alcohol by 100, line 167). While this makes the point about the difficulty using ABV to determine number of standard drinks, it may be easier and more intuitive for readers to provide a few example of drink types, volumes and ABV alongside the number of standard drinks.

• The objective outlined at the end (lines 225-7) may be too broad – while the impact of standard drinks labelling has implications for understanding consumption relative to health guidelines, the study’s examination of pouring does not address this directly.

Methods

• Was a sample size calculation conducted to inform the recruitment of 84 participants?

• This section could be edited in places to remove unnecessary details (e.g., type of measuring cylinder used to fill bottles, exact task wording for participants).

• Data collection (lines 325-8): could the authors clarify whether all participants completed the drink pours in the same random order, or whether order was randomised for each participant.

• As the 5%ABV beer container equalled one standard drink it was not possible to overpour a standard drink in this condition; therefore, it is difficult to compare the effect of labels on overpouring with other drink types.

• It would be useful to provide a link to the study protocol if this has been, or could be, uploaded to an open access repository.

Results

• Check for grammatical and typographical errors

• I think it would be helpful for readers if descriptive results were presented (e.g., mean and standard deviation) for the different measures across conditions.

• Exact p values should be given, rather than using ‘n.s.’

• A number of similar analyses and figures are presented – was there a prespecified analysis plan for the study? It may be clearer to present figure(s) for the primary analysis and move others to supplementary materials.

• It would be useful to make it clear in the data analyses section which is the primary outcome; explanations of the measures (e.g., reference to Weber’s Law) should be moved to the methods section too.

Discussion

• Details of the mean pour errors (lines 520-2) should be moved up to the results section.

• The assumption that drinkers would reduce their consumption in line with increasingly accurate standard drink pours (lines 522-7) may be too optimistic. While improved accuracy of pouring is valuable, and greater understanding of standard drinks may support people to monitor their alcohol consumption, it does not necessarily mean that they will monitor their drinking or change their overall consumption. There is evidence to suggest that, particularly young people, may use standard drinks to facilitate increased alcohol consumption and it is important to consider possible unintended consequences (e.g., JONES, S.C. and GREGORY, P. (2009), The impact of more visible standard drink labelling on youth alcohol consumption: Helping young people drink (ir)responsibly?. Drug and Alcohol Review, 28: 230-234. doi:10.1111/j.1465-3362.2008.00020.x)

• The results of the study should be put in the context of previous literature (which is absent in the discussion).

Data files: It would be helpful to have a data dictionary to help navigate the data

Reviewer #2: Clearly written and thoroughly analysed. The manuscript clearly illustrates the multiple steps currently required for individuals to estimate SDs. The public health benefit of SD labelling could also be enhanced if the authors new the % of all alcohol consumed from off-premise outlets, which relies of self-pouring compared to on-premise drinking which for liquors and wines are poured by bar staff. I would have liked to see a return to the discussion around legislation and policy in the Discussion and what policy implications the paper has, and whether any current move towards introducing SDs in the U.S.

Small errors:

- line 404 ‘There is an accuracy..’

-line 424 ‘categories of alcohol’

- line 446 ‘reasons’

6. PLOS authors have the option to publish the peer review history of their article (what does this mean?). If published, this will include your full peer review and any attached files.

Reviewer #1: No

Reviewer #2: No

---

## [Author Response · Author response to Decision Letter 0]

21 Sep 2020

Author responses are presented with [AU] preceding them. 

Reviewer comments are presented with reviewer number in front of them 

[AU] We were very pleased to hear that editors felt that “The paper makes an important contribution to the literature in terms of enhancing our understanding of beverage labelling strategies that can help improve the accuracy of pouring a standard drink (which could potentially make it easier for people to moderate their intake in relation to recommended guidelines).” - -- Additionally, at the suggestion of the editor, we have incorporated a limitations section into the document which was absent in the previous version. 

Journal Requirements:

2. Please ensure that you refer to Figure 7 in your text as, if accepted, production will need this reference to link the reader to the figure.

 [AU] This paper has undergone major streamlining and revision, which included the figures. To address this issue, we have conducted a careful read through of the document to make sure that all figures are appropriately acknowledged within the document text. There is no longer a Figure 7

3. Please upload a copy of Figure 9, to which you refer in your text by line 480. If the figure is no longer to be included as part of the submission please remove all reference to it within the text.

 [AU] Please refer to the comment immediately preceding this one. To address reviewer’s comments regarding the need for more streamlined prose and more concise manuscript, the entire document has gone through major revision and several figures have been removed. There is no longer a Figure 9.

Reviewers' comments:

Reviewer's Responses to Questions

Comments to the Author

1. Is the manuscript technically sound, and do the data support the conclusions?

Reviewer #1: No

Reviewer #2: Yes

2. Has the statistical analysis been performed appropriately and rigorously? 

Reviewer #1: Yes

Reviewer #2: Yes

3. Have the authors made all data underlying the findings in their manuscript fully available?

Reviewer #1: Yes

Reviewer #2: Yes

4. Is the manuscript presented in an intelligible fashion and written in standard English?

Reviewer #1: No

Reviewer #2: Yes

5. Review Comments to the Author

Reviewer #1: This is an interesting study that explores the impact of standard drink labels, compared to existing ABV labels, on accuracy of drink pouring across different types of alcoholic drink. However, the manuscript is very long and in places could be edited to reduce length and improve clarity. In addition, the stated objectives and discussion of results should reflect the outcomes examined in the study (i.e., pouring accuracy) rather than consumption, and the discussion of results should be placed within the context of previous literature. Outlined below are some specific comments that may be helpful for improving clarity. The manuscript should also be checked for grammatical and typographical errors.

Introduction

• [REVIEWER 1] The introduction is long and could be edited to make the points more succinct. For example, the first paragraphs provide quite a lot of detail of the current regulations around alcohol content labelling that could be summarised in a few sentences.

[AU]- We have SIGNIFICANTLY streamlined the manuscript, particularly the Background, Methods and Results, in light of this comment. The background no longer incorporates the depth of explanation regarding the historical and legal frame which undergirds alcohol labelling. The Methods have been significantly truncated to the most relevant points and the results now include only the raw pour data and a single variable that we perform analyses on, the one that is most germane to public/health policy (pour error as measured in standard drinks). We believe that this results in a document that is not only easier to read, also more easily interpreted for meaningful change. 

•[REVIEWER 1] The introduction could also be structured more clearly. For example, it would be useful to define ‘standard drinks’ (i.e., 14g pure alcohol in US – para. 7, line 147 and para. 8, line 163) and how these are used in drinking guidelines (Table 1) when they are first mentioned (para. 3, line 110). Followed by why standard drink labelling may be preferable to ABV information, given the wide variation both within (para. 6, line 136 and para. 9) and across drink categories and sizes (para. 7, line 150).

[AU] In the radically streamlined version of the introduction we have tried to do so, while emphasizing the disconnect between ABV labelling and the standard drink units used in health messaging. 

•[REVIEWER 1] Para. 8, lines 164-171: The explanation of the calculation is quite long and complex, and misses out some elements (e.g., dividing g of alcohol by 100, line 167). While this makes the point about the difficulty using ABV to determine number of standard drinks, it may be easier and more intuitive for readers to provide a few example of drink types, volumes and ABV alongside the number of standard drinks.

[AU] We agree that it is long. We have tried to streamline it a little, but the point we are trying to make is converting from ABV (which is what is present on current labels) to standard drinks (which are the units used in health messaging) is extremely difficult. We think showing how difficult makes this point well. 

• [REVIEWER 1] The objective outlined at the end (lines 225-7) may be too broad – while the impact of standard drinks labelling has implications for understanding consumption relative to health guidelines, the study’s examination of pouring does not address this directly.

[AU] Understood. We have worked throughout to reframe the language to indicate the ability to correctly identified a standard drink (which serves to inform consumption behavior). We have also revised the language such that it is more transparent in this regard. Further, the revision of the Results, which now only report a single dependent variable (pour error as measured by standard drinks) improves the document’s comprehensibility. Further, we now acknowledge in our limitations that consumption may actually be negatively impacted by imparting more accurate knowledge of alcohol content. 

Methods

• [REVIEWER 1] Was a sample size calculation conducted to inform the recruitment of 84 participants?

[AU]Power calculations which support the sample size have been incorporated into the beginning of the Methods section of the document. 

• This section could be edited in places to remove unnecessary details (e.g., type of measuring cylinder used to fill bottles, exact task wording for participants).

[AU] Understood. Much of the technical detail has been removed from this section. 

• [REVIEWER 1] Data collection (lines 325-8): could the authors clarify whether all participants completed the drink pours in the same random order, or whether order was randomised for each participant.

[AU] The randomization was conducted for the first set of trials for each individual participant. Replicants were performed in the same order as the first randomized set (which was (theoretically) unique to ever participant. So if participant ones randomization first set indicated the order of presentation to be 12, 1, 4, 3…. There second set (the replicates) were also 12, 1, 4, 3…. In other words, randomization was done by participant. We have modified the text within the Methods to enhance the clarity around this important point of process. 

• [REVIEWER 1] As the 5%ABV beer container equalled one standard drink it was not possible to overpour a standard drink in this condition; therefore, it is difficult to compare the effect of labels on overpouring with other drink types.

[AU]- This is correct. Although an ecologically valid condition (most low ABV beers cannot be overpoured in realistic conditions based on bottle (and glass) sizes), it does present an experimental imperfection. In an attempt to better address this, we have revised the language in the limitations to more directly address it (and also explain its presence).

• [REVIEWER 1] It would be useful to provide a link to the study protocol if this has been, or could be, uploaded to an open access repository.

[AU] We have taken advantage of the ability to upload Laboratory protocols in protocol.io as a supplemental material. This has not only allowed us to address the reviewer comments by streamlining the methods significantly, but also affords the opportunity to present the Methods with significant detail to interested parties. 

Results

• [REVIEWER 1] Check for grammatical and typographical errors

[AU]- We have tried to give the document a thorough read through to be sure that it is not only more concise, but that grammatical and typographical errors have been corrected throughout. 

•[REVIEWER 1] I think it would be helpful for readers if descriptive results were presented (e.g., mean and standard deviation) for the different measures across conditions.

[AU] Those data for the raw pour volumes are presented in Figure 2. For the standardized pour error variable they are presented in Figure 3.

• [REVIEWER 1] Exact p values should be given, rather than using ‘n.s.’

[AU] We have now done so, but note that the only time we used n.s., is when the calculated F was less than 1. The expected value for F under the null is 1, so statistical significance is impossible if F<1, thus there is a convention to report ns in those cases.

•[REVIEWER 1] A number of similar analyses and figures are presented – was there a prespecified analysis plan for the study? It may be clearer to present figure(s) for the primary analysis and move others to supplementary materials.

[AU] We revisited the Results upon receiving the reviewer’s comments and whole heartedly agree. We have removed redundant analyses and reworked the section significantly in an attempt to meaningfully synthesize the finding in a complete, yet more concise, form. We appreciate the feedback and believe that this change has significantly enhanced the readability of the document without altering the Conclusions that can be drawn from the data set. 

• [REVIEWER 1] It would be useful to make it clear in the data analyses section which is the primary outcome; explanations of the measures (e.g., reference to Weber’s Law) should be moved to the methods section too.

[AU] In streamlining the paper we have presented the error in standard drink units and have motivated doing so because those units are most relevant from a health perspective. This framing made the discussion of Weber’s Law superfluous and it has been removed. 

Discussion

•[REVIEWER 1] Details of the mean pour errors (lines 520-2) should be moved up to the results section.

[AU] With the new analyses this has been removed from the discussion.

• [REVIEWER 1] The assumption that drinkers would reduce their consumption in line with increasingly accurate standard drink pours (lines 522-7) may be too optimistic. While improved accuracy of pouring is valuable, and greater understanding of standard drinks may support people to monitor their alcohol consumption, it does not necessarily mean that they will monitor their drinking or change their overall consumption. There is evidence to suggest that, particularly young people, may use standard drinks to facilitate increased alcohol consumption and it is important to consider possible unintended consequences (e.g., JONES, S.C. and GREGORY, P. (2009), The impact of more visible standard drink labelling on youth alcohol consumption: Helping young people drink (ir)responsibly?. Drug and Alcohol Review, 28: 230-234. doi:10.1111/j.1465-3362.2008.00020.x)

[AU] Although making informed choices does depend on the ability to accurately assess the amount of alcohol in a standard drink, it is up to the drinker to use it. As the reviewer points out, this information could be used appropriately, ignored or used inappropriately. We have revised the language throughout the document to better reflect the complexities of human behavior and also worked to incorporate it into the state of knowledge specific to drinking behaviors. In addition, we explicitly acknowledge this point and reference the Jones & Gregory paper in the limitation section. 

• [REVIEWER 1] The results of the study should be put in the context of previous literature (which is absent in the discussion).

[AU] Please view the aforementioned comment. We now state that our results are generally in line with others who have examined this issue and cite that work. We also point out ways in which our approach was more nuanced than some of the prior work.

[REVIEWER 1] Data files: It would be helpful to have a data dictionary to help navigate the data

[AU] With our simplified reporting this might now be overkill, but we included a data dictionary in table 3.

• 

• 

[Reviewer #2]: Clearly written and thoroughly analysed. The manuscript clearly illustrates the multiple steps currently required for individuals to estimate SDs. The public health benefit of SD labelling could also be enhanced if the authors new the % of all alcohol consumed from off-premise outlets, which relies of self-pouring compared to on-premise drinking which for liquors and wines are poured by bar staff. I would have liked to see a return to the discussion around legislation and policy in the Discussion and what policy implications the paper has, and whether any current move towards introducing SDs in the U.S.

Small errors:

- line 404 ‘There is an accuracy..’

-line 424 ‘categories of alcohol’

- line 446 ‘reasons’

[AU] We appreciate that reviewer two believed our work to be “clearly written and thoroughly analysed”. That said, we have provided major revisions based on weakeness pointed out by Reviewer one. As part of those, we have reanalyzed our data such that the dependent variable is “normalized” so that errors are no longer volume based, but based on the # of standard drinks that one over or under pours. We believe that this not only addresses some of the concerns of Reviewer #1, but is also far more relevant to the desire that this reviewer has with regard to policy implications. Additionally, we have worked to revise the conclusions to focus more solidly on the same.

6. PLOS authors have the option to publish the peer review history of their article (what does this mean?). If published, this will include your full peer review and any attached files.

Do you want your identity to be public for this peer review? For information about this choice, including consent withdrawal, please see our Privacy Policy.

Reviewer #1: No

Reviewer #2: No

---

## [Editor Report · Decision Letter 1]

19 Oct 2020

Empirical evaluation of the presence of a label containing standard drinks on pour accuracy among US College students

PONE-D-20-14913R1

Dear Dr. Bix,

We’re pleased to inform you that your manuscript has been judged scientifically suitable for publication and will be formally accepted for publication once it meets all outstanding technical requirements.

Kind regards,

Victoria Manning

Academic Editor

PLOS ONE

Additional Editor Comments (optional):

Thank you for your careful consideration of the reviewers comments and for the thorough revisions made the manuscript which is substantially improved as a result.
---

## [Editor Report · Acceptance letter]

26 Oct 2020

PONE-D-20-14913R1 

Empirical evaluation of the presence of a label containing standard drinks on pour accuracy among US College students 

Dear Dr. Bix:

I'm pleased to inform you that your manuscript has been deemed suitable for publication in PLOS ONE. Congratulations! Your manuscript is now with our production department. 

Kind regards, 

on behalf of

Dr. Victoria Manning 

Academic Editor

PLOS ONE